# A Real-World Experience on the Efficacy of First-Line Treatment with Immune-Checkpoint Inhibitors in Non-Small-Cell Lung Cancer Patients with PD-L1 Expression ≥50%: The Role of *KRAS* Mutations

**DOI:** 10.3390/cancers17243980

**Published:** 2025-12-13

**Authors:** Lucia Motta, Samantha Epistolio, Jana Pankovics, Francesca Molinari, Benjamin Pedrazzini, Alexandra Valera, Luca Giudici, Stefania Freguia, Miriam Patella, Martina Imbimbo, Giovanna Schiavone, Milo Frattini, Patrizia Froesch

**Affiliations:** 1Oncological Institute of Southern Switzerland (IOSI), Ente Ospedaliero Cantonale (EOC), 6500 Bellinzona, Switzerland; lucia.motta@humanitascatania.it (L.M.); jana.pankovics@eoc.ch (J.P.); benjamin.pedrazzini@eoc.ch (B.P.); martina.imbimbo@eoc.ch (M.I.); giovanna.schiavone@eoc.ch (G.S.); 2Institute of Pathology, Ente Ospedaliero Cantonale (EOC), 6900 Locarno, Switzerland; samantha.epistolio@eoc.ch (S.E.); luca.giudici@eoc.ch (L.G.); stefania.freguia@eoc.ch (S.F.); milo.frattini@eoc.ch (M.F.); 3Thoracic Surgery, Ente Ospedaliero Cantonale (EOC), 6500 Bellinzona, Switzerland; miriam.patella@eoc.ch

**Keywords:** lung adenocarcinoma, KRAS, PD-L1, prognosis, survival, immune-checkpoint inhibitors

## Abstract

**Simple Summary:**

Lung cancer is the leading cause of cancer-related mortality all around the world. Despite advances in targeted treatment, mortality remains high with a survival rate that is less than 20%. The targeted therapies developed so far are specific for molecular alterations involving EGFR, HER2, ALK, and ROS1 genes; in the absence of these alterations, immunotherapies are administered. In previous years, therapies against KRAS (specifically KRAS p.G12C mutation) have also been developed. The aim of our study is to evaluate whether different KRAS mutation subtypes may play a role in modulating the response to immunotherapies in lung cancer. We described how advanced-stage patients harboring KRAS mutations demonstrated significantly longer median overall survival and progression-free survival compared to KRAS wild-type, and how progression-free survival was significantly shorter in patients with a KRAS p.G12D variant. Our data indicate how KRAS mutations, and specifically their subtypes, may modulate outcomes in advanced NSCLC patients treated with ICIs.

**Abstract:**

Background/Objectives: Several genetic alterations have been identified as drivers of uncontrolled cell growth in lung cancer, with KRAS mutations representing the most prevalent driver oncogene. Despite advances in targeted treatment, the 5-year survival rate of patients with advanced/metastatic NSCLC is still less than 20%. This study aims to assess the clinical relevance of KRAS mutations in the context of PD-L1 expression, focusing on patients with PD-L1 Tumor Proportion Score (TPS) ≥ 50% and treated with first-line immune checkpoint inhibitors (ICIs). Methods: We conducted a retrospective analysis of a real-world cohort comprising all staged NSCLC patients diagnosed and treated between 2018 and 2022 at our Institution with the available Next Generation Sequencing and PD-L1 immunohistochemistry results. Statistical analyses were made using the log-rank test, the two-tailed Fisher’s exact test, and Kaplan–Meier survival curves. Results: Among 520 NSCLC patients, 288 were adenocarcinoma (AC). Of these, 110/288 (38.2%) were KRAS mutants, and 83/278 (29.8%) presented a PD-L1 TPS ≥ 50%. In this subgroup, KRAS mutants demonstrated longer median overall survival (mOS) and progression-free survival (PFS) compared to the KRAS wild-type (28.7 vs. 10.7 months, *p* = 0.010; 6.4 vs. 3.5 months, *p* = 0.005, respectively). While OS did not differ among KRAS mutation subtypes, PFS was significantly shorter in patients with p.G12D (3.5 months, *p* = 0.03). Conclusion: This study is the first to investigate the interplay between KRAS mutations and PD-L1 expression in a real-world stage IV lung AC cohort treated with ICIs. Our findings indicate that the p.G12D mutation is associated with an extremely severe disease upon ICI monotherapy. These preliminary results need further validation in larger, prospective cohorts.

## 1. Introduction

Lung cancer is the second most frequently diagnosed cancer and is the leading cause of cancer-related mortality around the world. The percentage of new cases involving lung cancer is around 11%, with an estimated 1.8 million deaths per year [1]. For decades, traditional chemotherapy, radiotherapy, and surgery, in eligible cases, were the only types of treatment for lung cancer patients. Over the years, several genetic alterations have been identified as drivers of uncontrolled cell growth. Consequently, many target therapies and immunotherapies have been tested and contributed to improving treatment outcomes. Despite this, the 5-year survival rate is still less than 20%. Immune-checkpoint inhibitors (ICIs), either alone or combined with chemotherapy, have become the basis of treatment for patients with advanced/metastatic NSCLC not carrying driver mutations, and their use has also been shifted to neoadjuvant and adjuvant settings [2,3,4,5].

The Kirsten Rat Sarcoma Virus (*KRAS*) is the most frequently mutated oncogene found in non-small-cell lung cancer (NSCLC) (approximately 30% of all cases), with KRAS p.G12C being the most common mutation. KRAS is a small GTPase transduction protein involved in cell division and differentiation. This protein acts like a plasma membrane-localized molecular switch, regulating different transduction pathways such as PI3K-AKT-mTOR and RAF-MEK-ERK [6].

*KRAS* mutations predominantly occur in adenocarcinoma (AC) histology, in patients with a positive smoking history and Caucasian ethnicity, although it could be different depending on variant subtypes. About 39% of *KRAS* mutations are substitutions of glycine with cysteine at codon 12 (p.G12C), 21% are substitutions of glycine with valine (p.G12V), and 17% of glycine with aspartic acid (p.G12D). Several other mutations, generally located in codons 12, 13, and 61, occur at lower frequencies. While target therapies have been developed and approved for KRAS p.G12C mutations, such as Sotorasib and Adagrasib, therapies for KRAS p.G12D remain in the early stages of clinical development [7,8,9], including selective KRAS p.G12D inhibitors (MRTX113) and Pan-KRAS inhibitors [10,11,12].

Both KRAS p.G12C and KRAS p.G12D mutations are associated with worse clinical outcomes compared to *EGFR*-mutations in a large cohort of patients with stage I-III resected Lung AC from a French-Canadian retrospective study [13]. The clinic pathological characteristics and the prognostic significance of KRAS p.G12D need to be further investigated and could also define different clinical management [14,15,16]. It has been studied that the presence of KRAS p.G12C mutant tumors is correlated with an inflammatory tumor microenvironment and increased immunogenicity, with high PD-L1 expression and better potential sensitivity to ICIs [17,18]. To date, the primary treatment for stage IV NSCLC patients without actionable genomic alterations is based on PD-L1 expression levels, evaluated using the Tumor Proportion Score (TPS). Specifically, if PD-L1 expression is greater than or equal to 50%, ICIs like pembrolizumab monotherapy are among the first-line treatments used. Conversely, if PD-L1 levels are negative or between 1% and 49%, a platinum-doublet chemotherapy combined with ICIs like pembrolizumab can be administered [7,19,20].

Unfortunately, not all patients benefit from ICIs. It has been observed that only a subgroup of patients responds to ICI monotherapy, suggesting that other biomarkers may hinder immunotherapy efficiency. For example, it has been suggested that KRAS p.G12C and p.G12D mutations exhibit different clinical profiles and treatment responses, with patients harboring the KRAS p.G12D mutation showing less favorable outcomes with immunotherapy alone [16,21]. This preliminary data emphasizes the need to improve the knowledge of patients’ molecular profiles with respect to clinical outcomes, possibly leading to personalizing the treatment and enhancing clinical success rates. The aim of this study is to evaluate whether different *KRAS* mutation subtypes may play a role in modulating the response to ICIs monotherapy in NSCLC patients.

## 2. Materials and Methods

### 2.1. Patient Population

We retrospectively analyzed all consecutive patients diagnosed with non-squamous NSCLC (ns-NSCLC) at the Oncology Institute of Southern Switzerland (IOSI) between 1 January 2018 and 31 December 2022, regardless of disease stage at presentation. Treatment strategies were defined according to tumor stage, following multidisciplinary tumor board evaluation and in accordance with European clinical practice guidelines [7,19]. Clinical and pathological data were extracted from electronic health records. The dataset included demographic variables (age, sex), Eastern Cooperative Oncology Group Performance Status (ECOG PS) at diagnosis, body mass index (BMI), smoking status (never, former, or current smokers), key laboratory parameters, histological subtype, results of molecular profiling, tumor staging according to the eighth edition of the American Joint Committee on Cancer (AJCC), metastatic sites at diagnosis, treatment modalities administered, and follow-up outcomes (progression date, death, or last follow-up). Ethical approval for data collection and analysis was obtained from the local Ethics Committee (BASEC: 2023-01209; Ref. CE 4402; approval date: 13 July 2023). All participants provided written informed consent prior to data inclusion. Eligibility criteria comprised an age of 18 years or older at diagnosis, histologically confirmed ns-NSCLC, and the availability of clinical and molecular data. Patients were excluded if clinical information was incomplete, tumor tissue was insufficient for molecular testing, molecular results were inconclusive or not analyzable, or if informed consent was not obtained.

### 2.2. Treatment

All patients diagnosed with NSCLC (including advanced stage III and IV) were evaluated within a multidisciplinary tumor board (MTB) to determine the optimal treatment strategy. In general, systemic therapy was proposed to all stage IV patients with an Eastern Cooperative Oncology Group Performance Status (ECOG PS) of 0–2. First-line single-agent immunotherapy was administered to most patients whose tumors exhibited a PD-L1 TPS ≥ 50%. In selected cases, following multidisciplinary discussion, patients received a combination of ICIs and chemotherapy, or chemotherapy alone. For patients with a PD-L1 TPS between 1% and 49%, a combination of immunotherapy and platinum-based chemotherapy was recommended.

### 2.3. Molecular Analysis

Molecular analysis was conducted at the Institute of Pathology (ICP) EOC in Locarno, Switzerland. Point mutations and small insertions or deletions have been analyzed by NGS. For gene fusions, fluorescence in situ hybridization (FISH) was applied before 2020 and NGS in the following years. Genomic DNA was extracted following instructions from three 8 µm thick Formalin-Fixed and Paraffin-Embedded (FFPE) sections with the commercial kit QIAamp DNA FFPE Tissue kit (Qiagen, Chatsworth, Los Angeles, CA, USA). Next-Generation Sequencing (NGS) was performed on the extracted DNA using an Ion Torrent S5XL platform with the commercially available Ion AmpliSeq Colon and Lung Cancer Panel v2 (CLv2) (Thermo Fisher, Waltham, MA, USA). This panel gives information on the mutational status of 22 genes frequently mutated in lung AC. FISH analysis was performed on the FFPE tissue sections following already published criteria [22,23]. Starting from 2020, the Archer FusionPlex Lung NGS Panel (Archer, Boulder, CO, USA) was applied for detecting gene fusions in 14 genes (including *ALK*, *ROS1*, *RET*, *MET*, *NTRK1-2-3*). The methodology and interpretation criteria have already been described [24]. PD-L1 evaluation at the protein expression level was made using an automated instrument and the ready-to-use SP263 monoclonal rabbit anti-human antibody (Ventana Medical System, Tucson, AZ, USA) [25].

### 2.4. Statistical Analysis

Patient demographics and tumor characteristics were summarized using descriptive statistical methods, including calculations of central tendency and relative frequencies. Comparisons of continuous variables were performed using Fisher’s exact test (two-sided). Progression-Free Survival (PFS) was defined as the time from the first treatment to the date of clinical or radiological progression, while overall survival (OS) was the period from the date of histologically confirmed diagnosis to the date of death from any cause. Patients who were alive at the time of data analysis were censored at their last documented follow-up. Survival probabilities were estimated according to the Kaplan–Meier method, and differences between groups were evaluated with the log-rank test. A *p*-value below 0.05 was considered indicative of statistical significance. Statistical analyses were carried out using GraphPad Prism (version 9.5.1).

## 3. Results

### 3.1. Clinicopathological and Molecular Characteristics of the Cohort: Differences Between KRAS Wild-Type (WT) and KRAS-Mutated Cases

We identified 520 patients with ns-NSCLC who were consecutively diagnosed at our institution between January 2018 and December 2022. All patients underwent NGS to identify gene alterations. Among them, 288 patients were diagnosed at stage IV, of which a *KRAS* mutation was observed in 110 cases (38.2%). Clinical and demographic characteristics of this cohort are summarized in Table 1, based on *KRAS* mutational status.

In the *KRAS*-mutated population (n = 110, corresponding to 38.2% of the cases), the median age at diagnosis was 69 years (range: 45–86), with an almost equal distribution between males (n = 56) and females (n = 54). Most patients were current or former smokers. In these cases, at the time of diagnosis, brain metastases were present in 29/110 (26.4%) of patients, and bone metastasis in 35/110 (31.8%); in addition, PD-L1 expression analysis revealed that 68 patients had PD-L1 levels < 50%, while 39/107 (36.5%) patients had PD-L1 levels ≥ 50%.

Patients were categorized based on *KRAS* mutation subtype as illustrated in Figure 1.

In patients without *KRAS* mutations (*KRAS* WT) (n = 178, corresponding to 61.8% of advanced cases), the median age at diagnosis was 76 years (range: 31–100), with a higher proportion of males (102) than females (76). Most patients were current or former smokers, and approximately one-third were never-smokers. At the time of diagnosis, brain metastases were present in 54/178 (30.3%) patients, and bone metastases in 80/178 (44.9%). PD-L1 analysis in this group revealed that 127 patients had PD-L1 levels < 50%, while 44 had PD-L1 levels ≥ 50%, corresponding to 44/171 (25.7%).

### 3.2. Survival of the Patients with PD-L1 ≥ 50% Based on KRAS Mutation Status

We focused our attention on OS and PFS in the 50 patients with PD-L1 ≥ 50% who received pembrolizumab as first-line monotherapy (29 with *KRAS* mutations and 21 with a *KRAS* WT mutational status), in accordance with European guidelines. The remaining 33 patients, based on clinical judgment and multidisciplinary tumor board recommendations, were treated with chemotherapy alone, chemo-immunotherapy, or targeted therapy when an actionable molecular alteration was identified. Survival analysis (Figure 2A) showed that stage IV patients with PD-L1 ≥ 50% treated with pembrolizumab had a median OS (mOS) of 28.7 months in the *KRAS*-mutant group, compared to 10.7 months in the *KRAS* WT group (HR: 0.37; 95% CI: 0.20–0.68; *p* = 0.010).

PFS analysis following first-line pembrolizumab treatment (Figure 2B) demonstrated a significant difference between the two groups, with a median PFS (mPFS) of 6.36 months in *KRAS*-mutant patients versus 3.5 months in *KRAS* WT patients (HR: 0.33; 95% CI: 0.15–0.73; *p* = 0.005).

### 3.3. Survival of the Patients with PD-L1 ≥ 50% According to Specific KRAS Mutation Subtypes and Comparing the Subtypes to KRAS WT

We further analyzed OS and PFS according to specific *KRAS* mutation subtypes. No statistically significant differences in OS were observed among the various *KRAS* subtypes (*p* = 0.31) (Figure 3A). However, as shown in Figure 3B, patients harboring the KRAS p.G12D mutation (n = 4) had the shortest median PFS (mPFS) of 3.5 months, compared to those with p.G12C (mPFS = 19.8 months; n = 14), p.G12V (mPFS = 20.4 months; n = 7), and other *KRAS* mutations (mPFS = 6.3 months; n = 4). This difference was statistically significant (*p* = 0.03).

When comparing this same population to *KRAS* WT patients (Figure 4), we observed that *KRAS* WT patients had similarly poor outcomes, with an mOS of 8.6 months and mPFS of 3.5 months, comparable to the KRAS p.G12D subgroup. A statistically significant difference was observed for both OS (*p* = 0.01) and PFS (*p* = 0.001).

### 3.4. Survival of the Patients with PD-L1 < 50% Based on KRAS Mutation Status

Finally, we assessed OS and PFS in patients with PD-L1 < 50% who received chemo-immunotherapy. The *KRAS*-mutated population had a mOS of 23.9 months (Figure 5A), compared to 16.8 months in the *KRAS* WT group (HR: 0.72; 95% CI: 0.41–1.28; *p* = 0.26). PFS analysis in the same group of patients (Figure 5B) showed a mPFS of 6.9 months in *KRAS*-mutant patients versus 6.6 months in *KRAS* WT patients (HR: 0.89; 95% CI: 0.53–1.50; *p* = 0.62).

We also repeated the statistical analyses by stratification with *KRAS* mutation subtypes. In terms of OS (Figure 6A), there were no statistically significant differences among subgroups (*p* = 0.71), although a trend favoring the KRAS p.G12C subgroup (mOS = 47.3 months) was noted. Similarly, PFS analysis (Figure 6B) showed no significant differences between mutation subtypes (*p* = 0.84), with median PFS values ranging between 5 and 8 months across all groups.

## 4. Discussion

Our study, based on a real-world cohort of 288 stage IV patients treated and evaluated in a single institution, provides new insights into the prognostic and predictive role of *KRAS* mutations in advanced ns-NSCLC, particularly in the context of immunotherapy. We observed relevant clinical and molecular differences between patients harboring *KRAS* mutations and those with *KRAS* wt tumors. As expected, *KRAS* mutations were strongly associated with smoking history and occurred in a slightly younger population, whereas never-smokers were more prevalent in the *KRAS* WT group. These epidemiological features are consistent with previously reported data and support the role of *KRAS* mutations as a tobacco-associated oncogenic driver [19,26]. A key finding of our analysis is the improved survival outcomes observed in *KRAS*-mutant patients with PD-L1 ≥ 50% treated with first-line pembrolizumab monotherapy, compared with their *KRAS* WT counterparts: both OS and PFS were significantly longer in the *KRAS*-mutant cohort. Our data confirm the study of Torralvo and colleagues, who observed a superior response to ICI by *KRAS*-mutant patients in comparison to *KRAS* WT patients [27]. Our data therefore reinforce the notion that *KRAS* mutations may positively modulate tumor immunogenicity, possibly through higher tumor mutational burden or enhanced immune infiltration, thereby increasing responsiveness to immune checkpoint blockade [24,28,29].

When considering *KRAS* mutation subtypes, we found that patients with the KRAS p.G12D variant had notably poor outcomes after treatment with first-line ICI, especially with respect to KRAS p.G12C mutant cases. This observation was also reported by an Australian group, which reported that, at odds with patients carrying a KRAS p.G12C mutation, those with a KRAS p.G12D mutation did not display any survival benefit in the group that received immunotherapy or chemo-immunotherapy compared to chemotherapy alone [16]. The data demonstrating substantial resistance to immunotherapy in patients carrying the KRAS p.G12D mutation is currently explained by the evidence that KRAS pG12D mutant tumors are characterized by a distinct biological behavior, less favorable to immune-checkpoint inhibition [14,15,16,30]; indeed, KRAS p.G12D mutant tumors are characterized by a lower TMB [31] and by a lower proportion of cells of CD8+ T-cell infiltration, compared to non-p.G12D mutant cases [18]. Also in our cohort, although not reaching statistical significance due to the low number of patients, we observed a higher proportion of lymphocytic infiltration in KRAS p.G12C vs. KRAS p.G12D mutant cases. Furthermore, other explanations can involve structural and functional differences among KRAS amino acid substitutions, leading to different downstream signaling intensity and pathway preference [32]. Moreover, prior studies have reported a higher prevalence of STK11 and TP53 co-mutations in patients with KRAS p.G12D, both of which are well-known negative predictors of immunotherapy response. These alterations can reshape the tumor microenvironment by downregulating PD-L1 expression and reducing immune cell infiltration, particularly activated CD4 memory T cells, helper T cells, M1 macrophages, and NK cells [33]. However, it must also be mentioned that a report showing conflicting results, obtained by a Latin American cohort [16], indicated that further studies are recommended to shed light on this issue and that potential ethnic differences may be present even for *KRAS* mutations (and not only for EGFR mutations).

In contrast, patients harboring KRAS p.G12C or p.G12V mutations achieved the most favorable outcomes, particularly in terms of PFS. This finding may reflect intrinsic differences in the biology of these variants, including a greater propensity for high tumor mutational burden and enhanced neoantigen presentation, both of which are associated with improved responses to immune-checkpoint blockade [31,34]. Moreover, the clinical availability of KRAS p.G12C inhibitors presents additional therapeutic opportunities for this subgroup, with ongoing studies investigating their combination with PD-1/PD-L1 inhibitors [35]. Together, these data reinforce the concept that not all *KRAS* mutations carry the same predictive value for immunotherapy, and underline the importance of considering subtype-specific biology when designing treatment strategies [36,37]. Interestingly, in patients with PD-L1 < 50% treated with chemo-immunotherapy, no statistically significant differences were observed among *KRAS* subgroups, although a trend toward better outcomes for KRAS p.G12C was noted. This finding suggests that the combination of chemotherapy and immunotherapy may overcome the unfavorable prognostic impact of specific *KRAS* alterations, thus hypothesizing that in stage IV patients carrying a KRAS p.G12D mutation and with PD-L1 ≥ 50% the addition of chemotherapy to ICI monotherapy can be useful for patient treatment.

From a clinical perspective, our results reinforce the importance of integrating molecular profiling beyond PD-L1 expression to optimize treatment strategies. While PD-L1 remains a key biomarker for selecting immunotherapy, *KRAS* mutation status, particularly the different subtypes, may add prognostic and predictive information essential for a better prediction of ICI efficacy. The poor outcomes associated with KRAS p.G12D raise the question of whether these patients should be considered for alternative strategies, including clinical trials of novel agents. On the other hand, the favorable outcomes observed in KRAS p.G12C patients highlight the potential role of combining ICIs with direct KRAS p.G12C inhibitors.

Our study has some limitations. First, it is retrospective and single-center in design, which may introduce selection bias. Second, the number of patients in some *KRAS* subgroups, particularly in the one characterized by p.G12D, was small, limiting the robustness of statistical comparisons. Third, although we systematically assessed *TP53* and *STK11*, we did not evaluate *KEAP1*, because it was not included in the NGS panel applied for diagnostic purposes, even if it is another co-mutation known to influence immunotherapy outcomes. The main value of our study is that we are evaluating a real-world cohort; consequently, we have no bias from patients’ inclusion in clinical trials, and the data obtained are consistent with the preliminary data published in the literature, even in the subgroups with only a few cases.

## 5. Conclusions

To conclude, our data indicate that *KRAS* mutations, and specifically their subtypes, may modulate outcomes in advanced NSCLC patients treated with ICIs. KRAS p.G12C and p.G12V appear to be associated with favorable outcomes, whereas KRAS p.G12D patients show limited benefit from pembrolizumab monotherapy. These findings highlight the need for prospective validation and for clinical strategies that account for KRAS biology, including rational combinations of targeted therapy and immunotherapy.

## Figures and Tables

**Figure 1 cancers-17-03980-f001:**
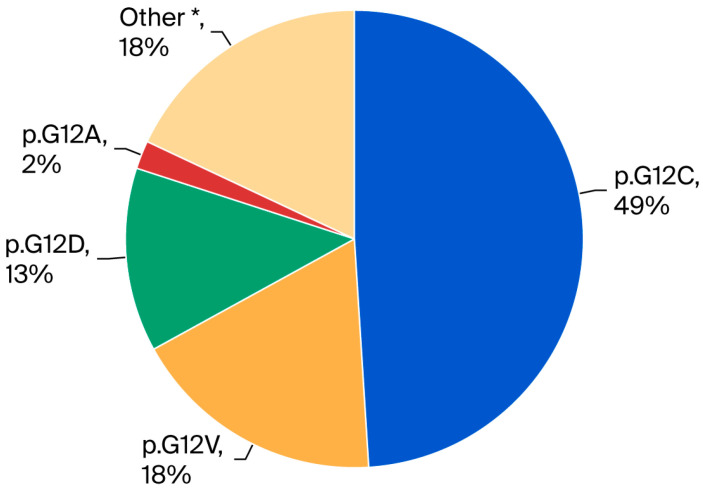
Pie chart showing *KRAS* mutation subtype distribution in stage IV ns-NSCLC (n = 110). The most frequent mutation was p.G12C (49%), followed by p.G12V (18%), other mutations (18%), p.G12D (13%), and p.G12A (2%). * Other: p.G12S, G12R, G12F, G13D, G13C, L19F, A146T, Q22K, Q61H, Q61L.

**Figure 2 cancers-17-03980-f002:**
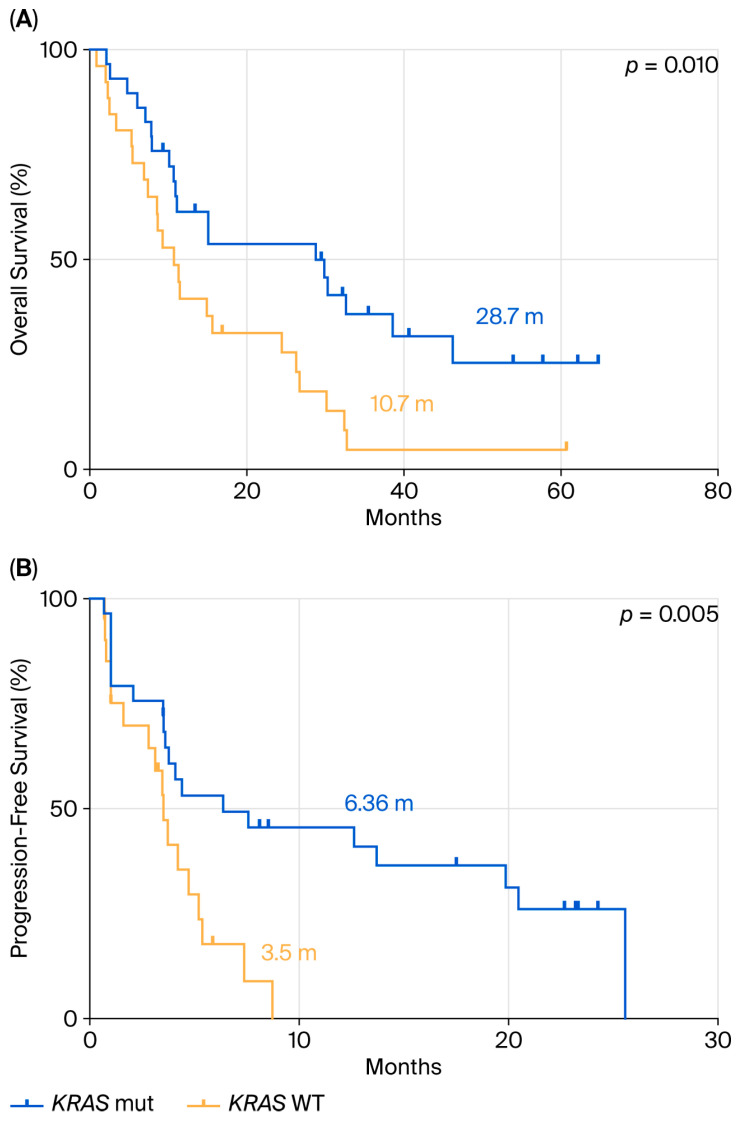
Comparison between *KRAS* mutant and WT in terms of OS (**A**) and PFS (**B**) in ns-NSCLC stage IV with PD-L1 ≥50% treated with pembrolizumab.

**Figure 3 cancers-17-03980-f003:**
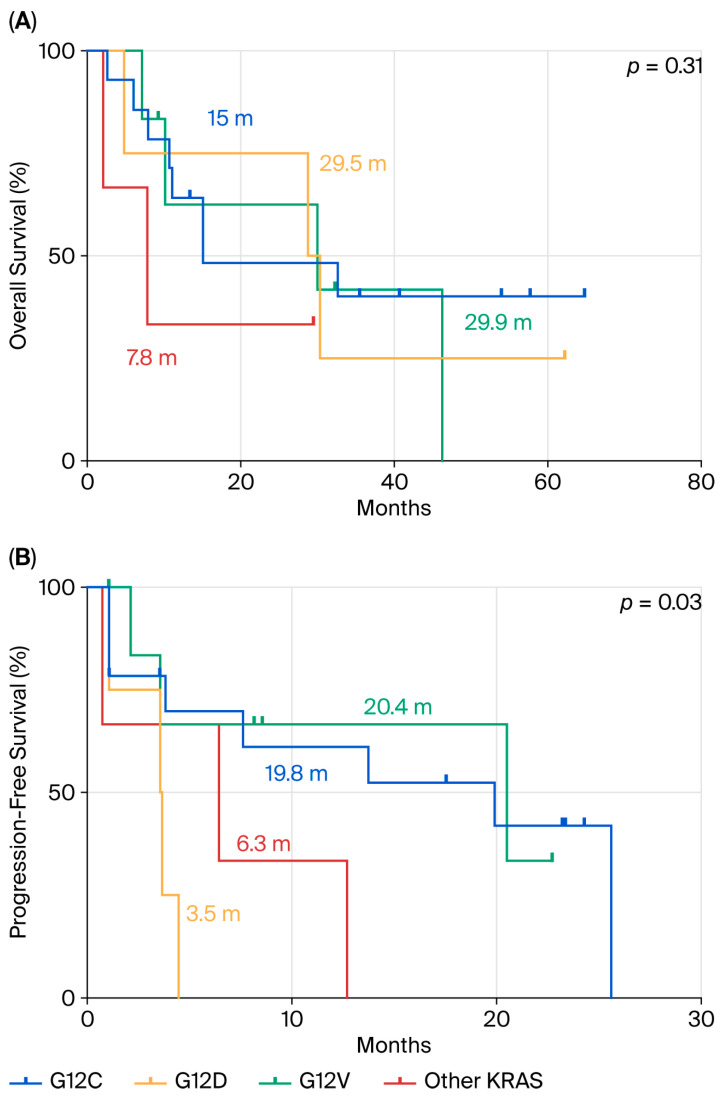
Comparison between p.G12D mutation and the other *KRAS* mutations in terms of OS (**A**) and PFS (**B**) in ns-NSCLC stage IV with PD-L1 ≥ 50% treated by pembrolizumab.

**Figure 4 cancers-17-03980-f004:**
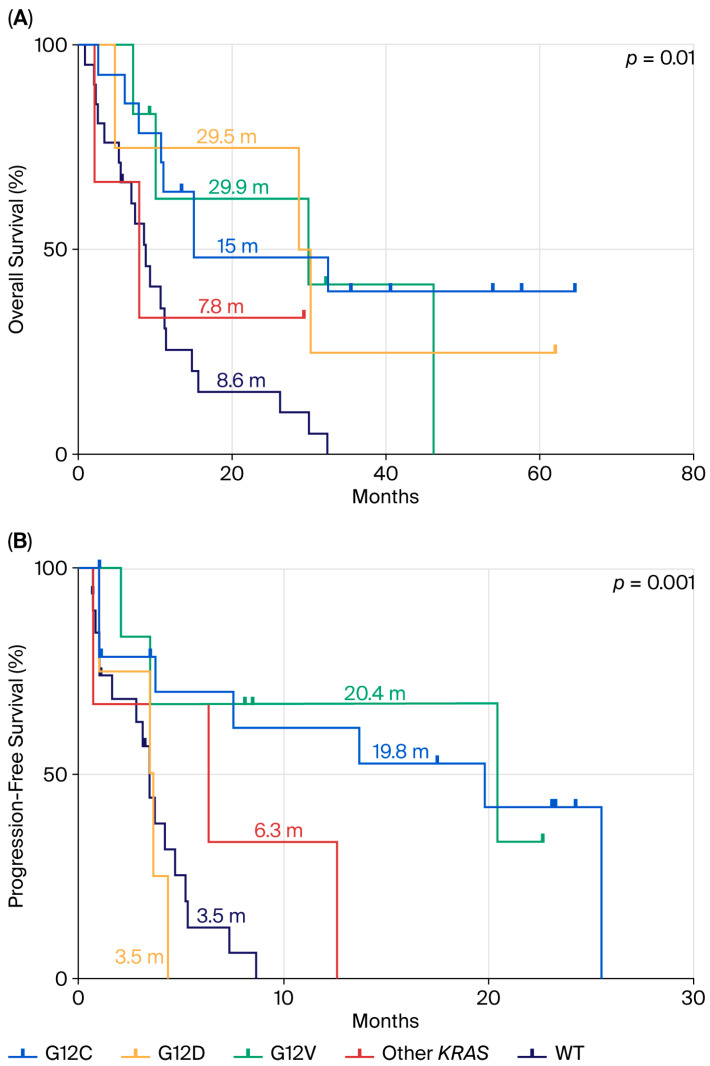
Comparison between *KRAS* WT population mutation and *KRAS*-mutated patients in terms of OS (**A**) and PFS (**B**) in ns-NSCLC stage IV with PD-L1 ≥ 50% treated by pembrolizumab.

**Figure 5 cancers-17-03980-f005:**
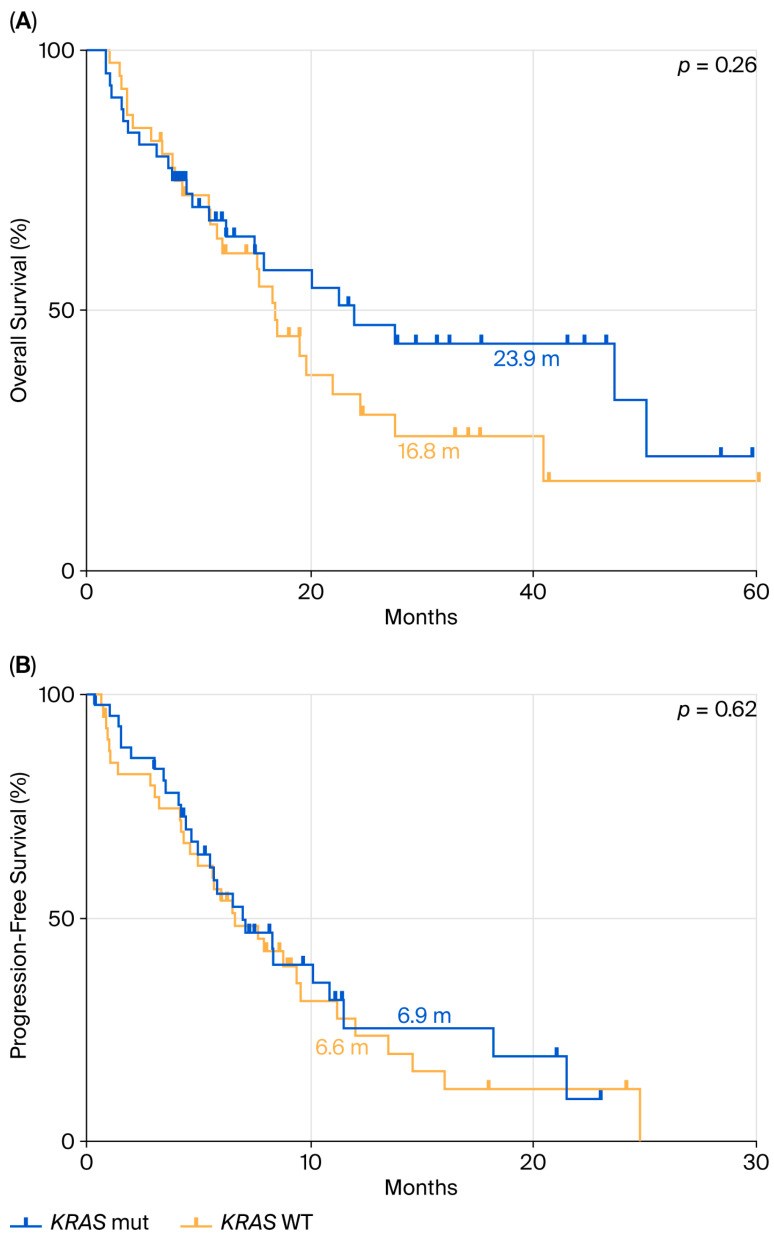
Comparison between KRAS mutant and WT in terms of OS (**A**) and PFS (**B**) in ns-NSCLC stage IV with PD-L1 < 50% treated with pembrolizumab.

**Figure 6 cancers-17-03980-f006:**
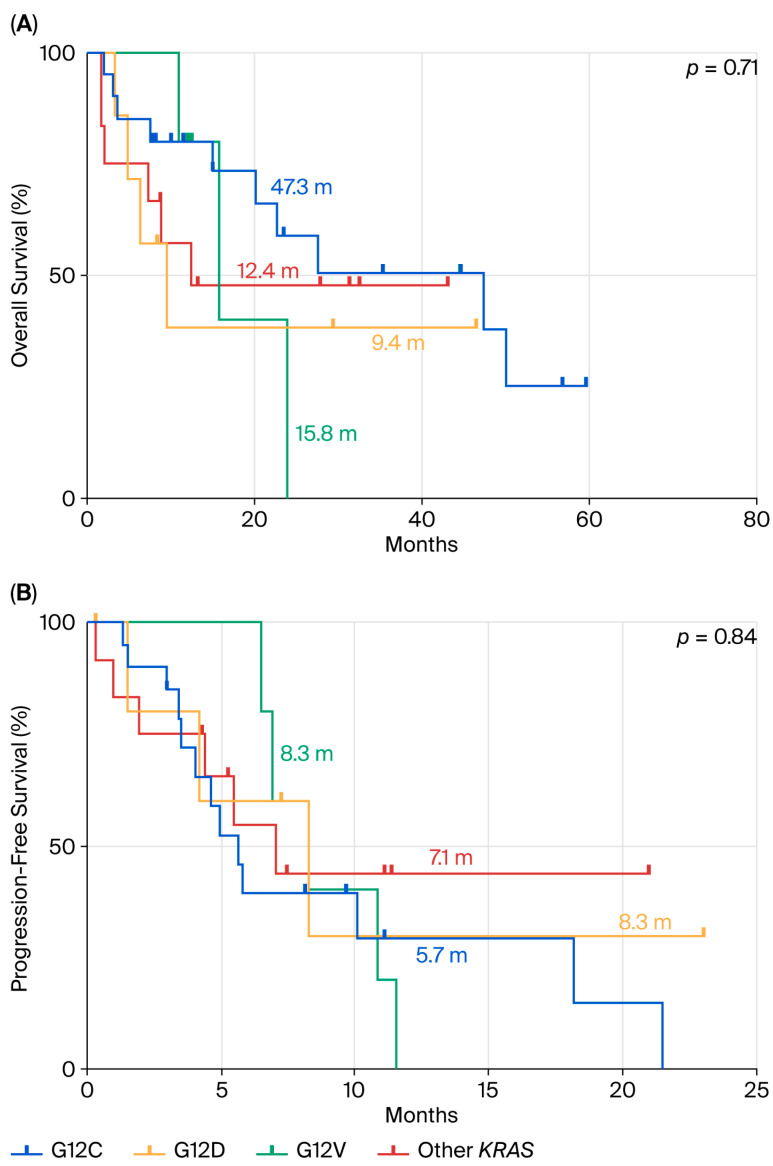
Comparison between p.*G12D* mutation and the other *KRAS* mutations in terms of OS (**A**) and PFS (**B**) in ns-NSCLC stage IV with PD-L1 < 50% treated by chemotherapy plus pembrolizumab.

**Table 1 cancers-17-03980-t001:** Clinical and demographic characteristics of stage IV ns-NSCLC patients categorized as *KRAS* wild-type (WT) or *KRAS*-mutated.

Stage IV Population	*KRAS*-Mutated	*KRAS* WT
Number of patients	110	178
Median age (range)	69 (45–86)	76 (31–100)
PD-L1 (<50%)	68	127
PD-L1 (not available)	3	7
PD-L1 (≥50%)	39	44
Of which 1st line Pembrolizumab	29	21
Sex (male/female)	56/54	102/76
Smoking habit		
Current smoker	66	68
Former smoker	33	52
Never smoker	6	48
Unknown	5	10
Brain metastasis	29	54
Bone metastasis	35	80
Mutation type		
p.G12C	54	
p.G12V	20	
p.G12D	14	
p.G12A	2	
Other *	20	-

* Includes p.G12S, G12R, G12F, G13D, G13C, L19F, A146T, Q22K, Q61H, Q61L.

## Data Availability

The datasets used and analyzed during the current study are available from the corresponding author upon reasonable request. The data are not publicly available due to institutional policy.

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
