# Peer review of "A Real-World Experience on the Efficacy of First-Line Treatment with Immune-Checkpoint Inhibitors in Non-Small-Cell Lung Cancer Patients with PD-L1 Expression ≥50%: The Role of KRAS Mutations"

_cancers, 2025, doi:10.3390/cancers17243980_

Round 1

Reviewer 1 Report

Comments and Suggestions for Authors

Dear Editor and Authors,

Thank you for asking me to review this manuscript titled "A real-world experience on the efficacy of first-line treatment with immune-checkpoint inhibitors in non-small cell lung cancer patients with PD-L1 expression ≥50%: the role of KRAS mutations." by Dr. Lucia Motta and her colleagues from Switzerland.

In this single institution, retrospective analysis the authors present their data from 288 with lung adenocarcinoma of which 110 (38%) also had a KRAS mutation and were treated with immunotherapy only if PDL-1 >50% or conventional chemo and immunotherapy if less. Overall survival, progression free survival where assessed considering KRAS mutation profile. The authors found that KRAS mutation improved overall and PFS in patients treated with immunotherapy alone, while the specific type of KRAS mutation present did not seem to play a significant role (possibly though due to the sample size!).

This is an interesting and poignant study. It is small in numbers, retrospective and from a single institution but it is well presented, used sound methodology and has interesting results. Thus it is valuable for publication if improved as there are a few issues that need addressing priot to. Please see my comments below.

Comments:

  1. I would suggest changing the sentence "For decades, 64 traditional chemotherapy and radiotherapy were the only type of treatment for lung can-65 cer patients." as it is inacurate. In eligible patients surgery is the gold standard treatment modality as I am sure the authors are well aware.
  2. The introduction is quite good but I suggest that the authors add some more information on the mechanism of action and function of the KRAS gene!
  3. PD-L1 TPS≥50% has long stopped been the cut off for immunotherapy as many studies has shown improved outomes with even lower % (as low as >1%!!). This needs to be addressed.
  4. Why was the period of study chosen to be from 2018 until 2022 and not after? Why the cut off?
  5. How complete where the electronic health records used to mine the data and how complete was follow up?
  6. Did the authors perform any type of sample size calculation or power analysis to assess if their number of patients used in the study were adequate to produce a statistical meaningful result?
  7. What stage was the cut off for "advanced NSCLC"? - line 131
  8. What is ns-NSCLC? Please define!
  9. The legend in Table 5 I think is wrong, should it not be PDL1<50% rather than >50%?

In conclusion, the study needs some major changes prior to it been published.

Kind regards.

Reviewer 2 Report

Comments and Suggestions for Authors

I think that the reader might be interested in this manuscript. However, there are some weak points. 1. There were only 4 patients in the KRAS p.G12D group, which was not enough for the research. 2. Could the author provide the information of next generation sequence of the paitient, and the TMB? 3. It might be better to analyze the microenvironments of the patients by IHC.  4. And I want to know that the efficacy of the treatment after of patients got resistance to ICIs.    

Comments on the Quality of English Language

It is better to further modify the English writing.

Reviewer 3 Report

Comments and Suggestions for Authors

I find the manuscript original and weel written, with an important analyses that could help future treatment's decision. 

The methods are correctly used and the results are clear and well presented, especially with the graphics. The subgroups analyses done and the stratification are well presented and supported. 

What I noticed is missed the role of Radiotherapy: 288 pts were Stage IV at diagnosis, 83 pts has brain metasasis at baseline. 110 pts of 288 were KRAS mut. We don't know the percentage of people with bone metastases (which we know are associated with poorer outcomes, worse PFS and OS). It's seems rather unlikely that none of these patients received RT at some point of their teatment, even for palliation or for OligoProgression or persistence. I think it would have been interestesd to notice how adding RT might affect the outcomes of the patients, in correlation with the KRAS status. I might suggest the authors to keep that in mind for the next analyses, if they decided to do a multicentric analysis or a prospective study. We don't know if the KRAS mutation (and which one in case) might modulate the immune response when RT is combined with IO or CT-IO.

The manuscript fits perfectly the journal’s scope
